# Seeing as Feeling? The Impact of Tactile Compensation Videos on Consumer Purchase Intention

**DOI:** 10.3390/bs14010050

**Published:** 2024-01-12

**Authors:** Kan Jiang, Shaohua Luo, Junyuan Zheng

**Affiliations:** 1School of Business, Guangxi University, Nanning 530004, China; 2102302024@st.gxu.edu.cn (S.L.); 2102302053@st.gxu.edu.cn (J.Z.); 2Key Laboratory of Interdisciplinary Science of Statistics and Management, Education Department of Guangxi, Guangxi University, Nanning 530004, China

**Keywords:** tactile compensation, perceived diagnosticity, mental imagery, solution innovativeness, consumer purchase intention

## Abstract

The lack of tactile experience is a significant flaw in online product evaluation and purchasing, but visual information can be utilized to compensate for tactile deficits. This study constructed a conceptual model based on mental imagery theory, innovativeness theory, and the personal goals framework, to explore the mechanism of visual–tactile compensation on consumer purchase intention. We conducted an online experiment with 406 participants recruited from a community and online store in Southern China and tested the research hypotheses using structural equation modeling. The findings suggest that visually compensated tactile perceived diagnosticity promotes mental imagery and sensory similarity, which, in turn, affects purchase intention. Theoretically, this study enriches the current explanations of online haptics by explaining the mechanisms by which haptic demonstration videos influence consumers’ haptic evaluations and behavioral responses, as well as the moderating role of personal goals therein; practically, this study offers advice for retailers seeking to build or expand their tactile marketing strategies.

## 1. Introduction

The sense of touch is one of the most prominent human senses [1]. In traditional offline purchasing activities, consumers always habitually touch the products in the store directly and feel their tactile characteristics (e.g., smoothness, pliability, etc.), to obtain a sensory experience from which they can infer product quality, as well as assess the value and make a purchasing decision [2,3]. Therefore, the sense of touch has an important impact on consumer decision-making and the marketing strategies of companies and retailers [4,5,6].

With the development of e-commerce, non-touch online shopping has become the mainstream consumption mode. However, the lack of haptic experience when purchasing online results from the inability of customers to touch the product, which increases the uncertainty of consumer evaluation and, consequently, leads to purchase refusal [5,7]. Two remedies have been proposed by the industry to address the lack of haptic experience and its negative consequences in online shopping. One approach is to interact with consumers through hardware that provides virtual touch functionality, thus simulating haptic feedback in place of real physical touch, so that consumers can have a similar sensory experience, which, in turn, affects their value assessment of the product [8,9]. For example, with VR wearables or AR apps, consumers can try on clothes, accessories, or try out furniture in a virtual environment for a more realistic shopping experience [10,11,12]. Another approach is the visually induced tactile compensation effect, where consumers mentally simulate tactile sensations based on tactile text descriptions, pictures, or video images [13,14]. For example, when selling clothes online, describing the softness of the product [6] or using tactile information and haptic imagery [15] can appropriately reduce decision barriers due to lack of exposure.

It is worth noting that, since hardware methods rely on the support of specific smart technologies, they cannot be widely applied due to technical difficulties and costs. Therefore, most retailers prefer relatively simple and economical visual compensation methods, i.e., tactile compensation is achieved by inducing consumers to generate virtual tactile sensations through the visual language of images [16,17], and product displays are enhanced by using images with descriptions of tactile information [15,18]. However, the tactile information conveyed by images is limited. Motion video can provide a richer sensory and value experience than static images [19]. It triggers consumers’ spontaneous perceptual reproduction by demonstrating effective tactile information and experiences, evoking memories of “real” touch, and manipulating imagined objects [20]. The ability of cues presented in product videos to effectively reproduce and convey haptic information depends on the diagnosticity of the haptic sensations perceived by consumers [14]. Perceived diagnosticity reflects the ability of a concept to convey information to aid in consumer assessment [21], and the more haptic cues and information a video presents, the easier it is for the customer to imagine that they are touching the product [22].

When consumers obtain sensory experiences by retrieving mental memories, they will imagine mental representations of stimuli and actions related to their past experiences and perceptual information at the time, a phenomenon known as mental imagery [23,24,25]. At the same time, people compare the received stimuli with the cues and make judgments according to the degree of perceptual fit between the cues [26,27], i.e., consumers retrieve information elicited and constructed from the cognition of mental simulations and compare it with their past experiences, which, in turn, facilitates decision making.

Research has shown that ambient tactile cues can fulfill consumers’ needs and influence their judgment of retailers’ capabilities [28]. Changes in retailing have shifted consumer preferences from value-oriented towards retailers that offer more functional advantages [29], and innovative functional solutions can provide higher satisfaction and value for rational and practical needs in consumption [30]. However, when examining how tactile cues influence consumer evaluations and attitudes in an online environment, the answer may be much more complex. Product demonstration videos trigger unique mental processes (e.g., mental imagery [17,31] and mental simulation [32]) that differ from other channels on multiple levels. For example, the pleasantness of the haptic experience may be less important as long as the mental imagery can be increased with similar (close) information [33]. This is crucial for the development of dynamic tactile online retail.

The previous study showed that goals activated by informational stimuli can influence decision-making [34]. Goal framing theory states that consumers will likely facilitate their behavioral choices by balancing perceptual and personal goals [35]. Enrichment goals drive consumers to pay more attention to the valuable information they acquire; on the other hand, when focusing on hedonic goals, consumers are more focused on their feelings and experiences than on the usefulness or otherwise of the information. Thus, personal goals influence their processing preferences for acquired information [6,36].

Given the mature potential of haptics in videos, individual consumer differences, and the lack of research on dynamic haptic displays, this study reveals the mechanisms influencing visual compensation strategies based on mental imagery theory, innovativeness theory, and goal-framing theory, aiming to explore three questions: (a) What is the role of visually compensated tactile perceived diagnosticity on mental imagery, sensory similarity, and consumer purchase intention? (b) Can programs that provide effective tactile cues in videos serve as a basis for retailers’ innovativeness judgments? (c) How do personal goals moderate consumer behavior? This study will attempt to provide retailers with insights into consumers’ psychological responses in order to design better tactile experiences and calibrate marketing strategies. Finally, this study provides retailers with guidance on displaying tactile cues in videos to promote positive brand response.

Therefore, this paper is organized as follows: Section 2 reviews the literature on visual–tactile compensation, combs through the concepts of perceived diagnosticity and mental imagery, and presents the hypothesis development. Section 3 explains the methodology of the paper. The data analysis and results are presented in Section 4. Section 5 presents the discussion and implications. Section 6 identifies the limitations of this study and directions for future research.

## 2. Literature Review and Hypotheses Development

### 2.1. Visual–Tactile Compensation and Perceived Diagnosticity

Visual–tactile compensation refers to the mental simulation of tactile contact that consumers evoke to imagine touching or using a product without physical contact [37]. Tactile compensation videos as stimuli tend to influence tactile judgments. For example, still images of the tactile properties of a product can activate mental imagery depicting the touch-related properties of clothing [38] and simulated touch experiences [32]. Additionally, observing the movement of a hand meaningfully touching a product can facilitate the haptic simulation experience and produce specific diagnostic effects [14].

Perceived diagnosticity refers to how consumers find a shopping experience helpful in evaluating products and making sound decisions. It positively contributes to consumers’ perceptions and evaluations of product attributes [21]. Perceived diagnosticity in the context of online displays helps consumers make informed decisions and positively affects their sentiment and purchase behavior [38,39]. Video solutions that use tactile demonstrations to showcase tactile features are more likely to convey tactile information or experiences related to the product than traditional flat-display videos. For example, by including demonstrations of hand or facial expressions or other choreographed object movements in the video to convey a sense of touch (silk fluttering in the wind represents the smoothness of the hair). Visual images can represent tactile sensations; the more vivid the image, the closer it is to actual perception [40,41]. This study introduced perceived diagnosticity into an online tactile context to measure the diagnostic ability of tactile perception conveyed by a tactile compensation video.

### 2.2. Mental Imagery

Mental imagery is an informative cognitive process of mental simulation created by a combination of recollection, entertainment, and stored information that evokes sensory representations and tactile perceptions in the consumer’s memory that resemble actual stimuli [42,43]. Despite being unable to touch a product directly, consumers can imagine the actual stimulus ‘in their minds’ [44]. Theories related to mental imagery have been widely used in marketing and consumer behavior, including online product presentations [17], virtual touch [45], and information processing [46]. According to the mental imagery theory, individuals use the information they perceive to mentally simulate sensory experiences to fill their inaccessible sensory information deficit and solve problems. Furthermore, stimulating consumers’ mental imagery captures their tactile imagery (including texture and smoothness), thereby influencing their virtual experience of the product [45]. This study adopts Liu et al.’s definition of mental imagery tactility [45], in which consumers imagine what they feel when touching, based on a video demonstration.

Visual stimuli can lead individuals to consider tactile elements and facilitate haptic imagery [44]. For example, visual languages such as text in vivid advertisements, travel advertisements, and images of online products can evoke mental imagery in consumers [45,47,48]. However, information quality can also affect mental imagery. This enhances consumers’ mental imagery if the scene image is perceived as fluent [15,17,48]. Conversely, visual stimuli that are difficult to perceive or that are contrary to expectations may inhibit mental imagery [48]. Additionally, because of the proactivity of touch, watching alternative touches can influence consumers’ product evaluations [14]. Similarly, if the tactile presentation conveys more tactile information, the greater the perceived diagnosticity of the tactile presentation, the more mental imagery is increased. Thus, this study proposes the following hypotheses:

**H1:** 
*The tactile perceived diagnosticity of visual compensation has a positive effect on mental imagery.*


Mental imagery is a crucial way of connecting information to consumers [20,49]. On the one hand, there is a positive relationship between mental imagery and purchase decisions; moreover, appropriate image processing can increase behavioral intentions [50]. Furthermore, by increasing the richness of images, composite images of motion and still images become more vivid than static images, for example, by increasing mental imagery and improving consumer preferences and purchase intentions [17]. On the other hand, images evoke mental imagery in consumers, giving them an analogous psychological experience similar to the real experience, thus facilitating purchase intentions [48]. Interactive moving images are more effective than static images at evoking mental imagery and adding value to the experience [38]. Virtual product experiences evoke tactile and spatial imagery and reduce product uncertainty [45]. Viewing videos of tactile demonstrations with more vivid and diagnosable information than pictures led to similar attitudinal changes. Thus, this study proposes the following hypothesis:

**H2:** 
*Mental imagery has a positive effect on purchase intentions.*


### 2.3. Sensory Similarity

Similarity is fundamental to an individual’s cognitive abilities in object recognition, recall, and problem-solving. Therefore, assessing the similarity of sensory cues is a vital information-processing step [51]. Sensory similarity refers to the extent to which a product’s sensory experience mimics its physical and sensory experiences, helping consumers relate the presented tactile cues to the actual touch substitution at the perceptual level and obtain a relevant sensory experience [27]. Additionally, sensory similarity requires a ‘fit’ between the depicted haptic cues and the substitution sensation, which supports the receiver in receiving similar tactile experiences and thus filling sensory gaps [27,52]. This study defines sensory similarity as the degree of similarity between memories or alternative tactile stimuli that consumers search for as substitutes based on video presentations.

Clear and vivid product descriptions are accompanied by sensory stimulation and activation [27]. When consumers are unsure of the sensations depicted, they can use realistic and accessible objects to replace online tactile sensations and increase their perception of similarity [53]. By increasing the representation of the tactile properties of products and demonstrating them, consumers are given more opportunities for analogies. For example, adding specific, consistent, or similar cues to traditional shops can increase consumers’ dwell time and boost their likelihood of purchasing [19,27]. Furthermore, similar sensory perceptions improve consumer attitudes and promote purchase intentions [3,54]. Therefore, this study suggests that consumer assessment of sensory similarity can be improved by enhancing their perceived diagnosticity of tactile stimuli in tactile presentations or haptic representations between stimuli and products. Thus, we propose the following hypothesis:

**H3:** 
*The tactile perceived diagnosticity of visual compensation positively influences sensory similarity.*


**H4:** 
*Sensory similarity significantly affects purchase intention.*


### 2.4. Solution Innovativeness

Innovation theory summarizes innovativeness as a broad range of outcomes of a firm’s activities and is an ongoing assessment of a firm’s innovative capabilities from a consumer perspective [55,56]. At the macro level, firm innovativeness is a consumer’s perception of enduring firm capabilities that bring innovative ideas and solutions to the market [55,57]. At the micro level, perceived retailer innovativeness improves consumer perceived value and increases loyalty and behavioral intentions [58,59]. Most studies on innovativeness emphasize that retailers’ innovation activities should be oriented toward consumer-perceived solutions. Stock et al. define the concept of solution innovativeness as follows: ‘solution innovativeness combines the practicality and novelty of the solution created’ [60,61]. Accordingly, this study defines solution innovativeness as a retailer’s ability to design novel shopping solutions that meet consumers’ practical needs.

Providing ways to facilitate consumer assessment of tactile attributes is an important manifestation of innovation for online retailers [28]. First, providing vivid tactile demonstrations helps consumers simulate virtual mental experiences and improve their image of product consumption [17]. Second, mental imagery influences product memory and triggers experiential consequences and positive evaluations [47]. Providing methods that facilitate tactile assessment may enhance consumer evaluations of retailers’ innovation capabilities.

Little research has been conducted on the correlation between sensory similarity and solution innovativeness, but some evidence exists. Similarity ratings represent consumer feelings and emotional experiences [51]. Previous research has highlighted that the more similar a stimulus feels to consumers, the less information they seek outwardly and the shorter their information processing, improving their evaluation of the brand [27,62,63]. Similarly, this study expects that positive mental imagery and sensory similarity ratings lead consumers to feel the effectiveness of tactile reach. This will lead to confidence in decision-making and increase positive evaluations of the retailer’s ability to create new solutions. Accordingly, we hypothesize the following:

**H5:** 
*(a) Mental imagery and (b) sensory similarity significantly affect solution innovativeness.*


Solution innovativeness represents a retailer’s ability to offer an effective solution and encompasses consumers’ evaluation of characteristics such as novelty, usefulness, and effectiveness [60,61]. When a solution is perceived as helpful, consumers change their attitude toward it and consider it positive [64]. For example, perceived innovativeness in a retailer’s ability to provide innovative services leads to consumer satisfaction and loyalty, influencing consumer patronage decisions and making them more likely to shop with the retailer [65,66,67]. In other words, when consumers perceive a particular approach as a practical solution, they are particularly likely to rely on providing that solution and ultimately generate purchasing behavior [68]. Additionally, the emotional satisfaction of consumers’ perception that providing effective solutions is an innovative effort by the retailer to make things easier for them will promote positive behavior [59]. Therefore, this study proposes the following hypotheses:

**H6:** 
*Solution innovativeness has a significant effect on purchase intentions.*


### 2.5. The Moderating Role of Personal Goals

Personal goals represent an individual’s desired state and are drivers that influence decision-making [34,69]. According to the goal framework theory, goals ‘construct’ the way people process information and take action [35]. The goal framework includes gain, hedonic, and normative goals. Consumers with gain goals will likely be more sensitive to the resources, value (e.g., quality and functionality), and product cues they receive. While consumers with hedonic goals are more likely to be influenced by pleasure, mood changes, and so on, normative goals cause consumers to follow personal and social norms and behave ethically [34]. Consumer goals and motivations play important roles in evaluating different stimuli and decisions [70]. Given the online purchasing environment, the tactile assessment of a product is not related to its social dimension; therefore, only gain and hedonic goals are discussed in this study.

Differences in personal goals influence consumers’ emotional and cue dependence, thereby affecting their behavior [70]. On the one hand, when mental imagery is low, consumers usually rely more on the visual information on the screen. At this point, consumers may notice cues relevant to the product and consumer interaction (e.g., presentation style) [8]. Lighthearted graphics can provide an increasingly pleasant emotional experience, thus improving the quality of interactions and emotional connections [71]. However, owing to increased mental imagery levels, consumers expect effective information to improve task efficiency [45]. Therefore, gain-goal-directed individuals perform mental simulations to obtain instrumental tactile information through object haptic cues in tactile compensation videos to quickly identify tactile attributes, extract tactile information, and facilitate purchase intentions [13]. In other words, the guidance of consumer mental imagery must be related to assessing product value to reduce uncertainty and thus generate a better shopping experience. Gain goals are crucial for driving consumers to pursue outcomes and achieve task efficiency, and the effectiveness of solutions is often measured regarding value assessment [60,61]. Thus, when mental imagery levels are high, achieving goals drives consumers to evaluate products rationally through interpretive elements and conform to their expectations of product value, thereby reducing uncertainty. Accordingly, the following hypothesis was formulated:

**H7:** 
*At lower levels of mental imagery, hedonic goals enhance the facilitative effect of mental imagery on purchase intention; at higher levels of mental imagery, gain goals enhance the facilitative effect of mental imagery on purchase intention.*


The assessment of sensory similarity involves processing perceptual tactile information, and the target affects preferences for extracting and using the acquired tactile information [6,27]. When sensory similarity ratings are low, consumers may focus more on the practical value of video delivery and expect information to be captured or retrieved by stimulating the same sensory modality. Under limited tactile sensory similarity conditions, motivations such as hedonic pleasure are subconsciously weakened, and information processing focuses on rational value choices. Hedonic consumers expect to feel better about themselves and rely on their sensations and emotions [35]. In an online environment, consumers with hedonic goals may be attracted to products if their images have similar sensory perceptions. At higher levels of sensory similarity, consumers receive stimulating experiences from the real and image-driven senses. They explore hedonic information to enhance enjoyable shopping experiences through alternative tactile experiences [50]. Thus, a high sensory similarity is more likely to allow hedonic goal-oriented consumers to experience pleasure from videos. Accordingly, we assume the following:

**H8:** 
*At lower levels of sensory similarity, gain goals enhance the facilitative effect of sensory similarity on purchase intention. At higher levels of sensory similarity, hedonic goals may improve the contribution of sensory similarity to purchase intention.*


The theoretical framework of this study is as shown in Figure 1:

## 3. Methodology

### 3.1. Measurement

This study uses variables sourced from well-established scales. Perceived diagnosticity was measured by adapting items from [10]; solution innovativeness was adapted from [10]; mental imagery was measured according to [72]; sensory similarity was adapted from [27]; and consumer purchase intention was measured by [73]. All items are in English, and we used the standard ‘translation-back translation’ procedure to finalize the Chinese version for the Chinese respondents. The questionnaire was revised using a pretest (*N* = 75) to ensure the items were clear and easy to understand. A 5-point Likert scale (1 = ‘totally disagree’, 5 = ‘totally agree’) was used for all items. Table 1 shows the final items.

Additionally, we used the meaning of secondary and potential goals from the Consumer Motivation Scale to interpret gain and hedonic goals. The participants were asked to make their goal choices accordingly. For example, the goal grouping asked, ‘When examining this product, my consumption goal is more in favor of ......’ with options (2 = hedonic, 1 = gain). Participants were divided into two groups based on their scores for gain and hedonic goals (Table 2).

### 3.2. Pretest

We selected clothing as the experimental material because garments and fabrics are considered to have visual and tactile properties [38]. Based on this, we conducted interviews with consumers who have experienced online shopping to further confirm the experimental materials. During the interviews, we asked participants about the types of clothing/fabrics they frequently purchased online and whether they used a particular clothing/fabric regularly. The results showed that more than 80% of the women chose chiffon dresses while more than 85% of the men chose cotton shirts and reported that they could perceive visual and tactile characteristics from them. Therefore, we chose chiffon dresses and cotton shirts as experiment materials.

Specifically, the tile display video of the garment contains information on appearance attributes such as color, size, design, and fabric. The tactile compensation videos show the model wearing the garment and demonstrate the tactile sensations and supporting items. Seventy-five subjects were recruited through the ‘Sojump’ platform. Participants were informed that the experiment aimed to identify the video type. Finally, the subjects were asked to rate the two videos (1 = tiled display video and 5 = tactile compensation video) according to the definition of tiled mons. The lower the score, the more the participant considered the video to be a flat display and vice versa for a tactile compensation video.

The questionnaire was valid for all 75 participants. Independent-sample *t*-tests indicated that tactile compensation videos scored higher, on average, than tiled display videos (*M*_tiled_ = 1.45; *M*_tactile_ = 4.57, *t* = −14.94, *p* < 0.001). The results support the identification of the type of tactile compensation video.

### 3.3. Materials and Data Collection

This study used an online questionnaire to collect data. The questionnaire included video recall stimuli, target selection, measurement of question items, and validity tests of experimental manipulation. In addition to the main structure, the questionnaire incorporated demographics, such as consumer gender, age, education level, and frequency of online purchases. The survey was awarded RMB 3 per participant to ensure the quantity and quality of the questionnaire.

The participants were consumers from various industries recruited online and offline between January and March 2023. After starting the experiment, the researcher sent appropriate product links to the subjects’ mobile devices for free browsing through the product information. In addition to the basic information, two videos were included (tiled display and haptic demonstration), which were otherwise consistent (e.g., duration and background) and excluded the influence of information such as branding. Finally, an experimental questionnaire was distributed, and the entire experiment lasted for approximately 15 min per subject. A total of 629 questionnaires were received, and 406 valid responses were obtained after the screening. The participants were 52% female and 48% male, with an average age of approximately 25 years, and more than 78% had a bachelor’s degree (Table 3).

## 4. Data Analysis and Results

### 4.1. Reliability and Validity Tests

SPSS (version 27.0) and AMOS (version 26.0) were used to verify the model’s fit, reliability, and validity. Specifically, all indicators met the model fit criteria (χ^2^/df = 1.331 < 3, root mean square error of approximation = 0.029 < 0.08, comparative fit index = 0.985 > 0.9, Tucker–Lewis index = 0.981 > 0.9, and normalized fit index = 0.942 > 0.9). Additionally, the results of the factor analysis (Table 4) showed that Cronbach’s α and the combined reliability values for each factor were greater than 0.7, and the average extracted variance (AVE) value was greater than 0.5, indicating good convergence. The AVE square root values for each variable were greater than the absolute values of their correlation coefficients, demonstrating the good discriminant validity of the variables (Table 5).

### 4.2. Model Hypothesis Testing

We used AMOS 26.0 to validate the main effects. The results showed (Table 6) that perceived diagnosticity had a significant positive impact on mental imagery and sensory similarity (H1: β = 0.554, *p* < 0.001; H3: β = 0.406, *p* < 0.001). Mental imagery and sensory similarity significantly influenced consumers’ perceived solution innovativeness and purchase intention (H2: β = 0.214, *p* < 0.001; H4: β = 0.182, *p* = 0.022; H5a: β = 0.19, *p* < 0.001; H5b: β = 0.299, *p* < 0.001). Solution innovativeness significantly influenced purchase intention (H6: β = 0.397, *p* < 0.001). The results suggest that the perceived diagnosticity of tactile stimuli has a direct positive impact on mental imagery and sensory similarity and facilitates purchase intention through solution innovativeness. All H1 to H6 were confirmed to be supported (Figure 2).

### 4.3. Moderating Effects

Hierarchical regression analysis was used to examine the moderating effects of personal goals. The results of the analysis of the moderating effect of the pathway of mental imagery to purchase intention (Table 7) showed that three variables had a significant impact on purchase intention [R^2^ = 0.184, F (3, 402) = 30.267, *p* < 0.001]. Specifically, personal goals significantly moderated the relationship between mental imagery and purchase intentions [B = 0.195, *t* (406) = 2.026, *p* = 0.043]; gain goals contributed more to the effect of mental imagery on purchase intentions than hedonic goals. A simple slope plot (Figure 3) reveals the magnitude of the difference in the impact of individual goals on mental imagery, which promotes purchase intention. The results show that, at lower levels of mental imagery, hedonic goals positively moderate the positive effect of mental imagery on purchase intentions. At higher levels of mental imagery, the gain goal positively moderated the positive effect of mental imagery on purchase intention. Therefore, H7 holds.

Table 8 presents the results of the hierarchical regression analysis of the sensory similarity to the purchase intention pathway. There was no moderating effect of personal goals on the relationship between sensory similarity and purchase intentions [R^2^ = 0.1082, F(3, 402) = 13.091, *p* = 0.65], implying that personal goals did not affect the relationship between sensory similarity and purchase intention. Therefore, hypothesis H8 does not hold.

## 5. Discussion and Implications

### 5.1. Discussion

This study aimed to explore the mechanisms by which tactile sensations compensated by visuals influence consumers’ purchase intentions. First, the results of the SEM analysis suggest that visual compensation based on tactile compensation videos increases mental imagery and sensory similarity through tactile perceived diagnosticity, which, in turn, increases consumers’ perception of the innovativeness of the solution and, ultimately, positively influences purchase intentions. This study introduces perceived diagnosticity as a measure of the effectiveness of tactile reach and elucidates the mechanisms by which demonstration videos influence consumers in terms of tactile ‘reach’.

Second, similarly to previous research, innovative and effective solutions enhance consumers’ perceptions of retailers’ abilities to provide innovative solutions [76]. Specifically, online retailers offering solutions that help consumers understand and evaluate tactile attributes can improve the shopping process and reduce uncertainty. Consumers perceive retailers’ efforts and translate them into positive behavioral intentions. The experiment showed that the perceived diagnosticity of tactile stimuli enhances consumers’ perceived novelty and the utility of retailers’ effective solutions by activating mental imagery and sensory similarity. This study confirms that innovativeness can be considered an extrinsic cue to assess retailer perceptions from an online shopping perspective, influencing purchase intentions through ‘functional–cognitive’ and ‘affective–experiential’ processing pathways [28,55]. We found that enhanced mental imagery and sensory similarity assessments helped consumers make decisions when sensory-rich tactile compensation videos were provided to demonstrate tactile information. As a result, consumers can perceive the retailer’s efforts to optimize their shopping experience, which, in turn, leads to positive evaluations. These findings highlight the relevance and importance of the ability to deliver haptic cues in shaping innovativeness.

Third, this study discusses the moderating role of personal goals on purchase intention. The results showed that, when the level of mental imagery was low, hedonic goals promoted the positive influence of mental imagery on purchase intention. When the level of mental imagery is high, achieving goals can boost the positive influence of mental imagery on purchase intention. This may be because customers appreciate and approve of the solutions offered by retailers, making them more likely to buy. The theory of planned behavior states that, when consumers focus on achieving goals, the rational choice process allows them to plan to motivate behavior according to their goals; thus, they prefer elements with value [77,78]. Additionally, personal goals did not differ significantly in the pathways through which sensory similarity influenced purchase intention. Embodied cognition theory explains that cognition is derived from physical a priori knowledge. Therefore, similarity perceptions without physical contact are not affected by personal goals [79]. A practical tactile approach can induce consumers to clarify goals and reinforce values and emotional experiences, providing practical support for designing tactile presentation videos.

### 5.2. Theoretical Implications

Based on mental imagery and innovativeness theories, this study explains the mechanism of tactile influence on consumer behavior compensated by vision, enriching the visual–tactile literature with certain theoretical contributions.

First, it extends the research on visually induced tactile compensation effects in online retailing and the visual–tactile interaction framework [26] while enriching the sensory marketing framework theory [5]. Unusually, this study further revealed changes in personal information processing, constructing explanations for the mechanisms at play in purchase intentions through perceived diagnosticity, mental imagery, and sensory similarity. This study investigated the impact of the tactile effects of visual compensation on consumers by developing a theoretical framework. Previous online tactile research focused on comparing the outcomes of different types of haptic reach, with little attention paid to the measurement of tactile diagnosticity. Moreover, it has chiefly used concepts such as perceived usefulness and perceived informativeness to explain the consequences of haptics that are dependent on specific technologies. This study bridges the gap between visually induced tactile compensation and measuring reach effectiveness by introducing perceived diagnosticity to calibrate the ability of an offered solution to convey tactile information and be perceived by consumers. Overall, in the visual–tactile compensation effect, a consumer’s psychological change process goes through cognition–emotion/emotion–behavior, where the cognitive process includes changes in information processing to evaluate perceived diagnosticity, mental imagery, and sensory similarity.

Second, this study considers retailers’ innovative capabilities. We extend our understanding of the role of haptic cues in shaping judgments of retailers’ innovativeness in online retail service environments. This is consistent with traditional tactile research, where haptic cues can underpin assessments of retailers’ capabilities [28]. We examine the ability of retailers to design shopping solutions that meet consumers’ practical needs—that is, solution innovativeness—focusing on whether the innovative form of design solves practical problems and optimizes shopping efficiency [61]. Suppose that a retailer provides a tactile demonstration that effectively addresses tactile deficits. In this case, customers will not only experience the diagnostic effect but will also be prompted to develop a positive opinion of the retailer, implying that consumers’ assessment of haptic cues in the online retail environment will likely be based on the evaluation of the retailer’s capabilities. The premise of this study’s findings is that a retailer’s performance can be used as a reference point, allowing consumers to make relative diagnoses of haptic cues related to the retailer’s capabilities and demonstrating the spillover effect of messaging. Effective tactile compensation solutions enhance consumers’ perceptions of retailers’ abilities to deliver innovative solutions [28,76]. This study’s exploration of innovativeness advances the research on the impact of online tactility on perceived retailer innovativeness and its subsequent effects, adding new insights to the existing literature on retailer innovativeness.

Third, we discuss the influence of personal goals on consumer behavior. Unlike previous studies that suggest that ‘individual differences in touch needs are accompanied by different levels of sensory stimulation and activation’ [15], this study explains the borderline role of personal goals on purchase intentions from the consumer motivation perspective. Specifically, when mental imagery is high, achieving goals can enhance the facilitative effect of mental imagery on purchase intention. This is because gain goals are essential factors in driving behavior. This psychological motivation increases the rational choice process of consumers, and individuals choose products in ways that satisfy their goals [80]. The process of receiving information is highly correlated with consumer goals, so different goals drive consumers to think about decisions from different paths. Furthermore, hedonic goals did not affect the relationship between sensory similarity and purchase intention. Zagan’s cognitive theory states that physical foundations determine an individual’s cognition in terms of embodied cognition. The development and expression of cognition are limited by the embodiment of the cognitive subject and the various contextual factors in which it is immersed [81]. Understanding the differences in the goals and motivations of consumers to influence their choices and preferences is essential for retailers to develop practical initiatives that promote purchase behavior.

### 5.3. Management Implications

First, online retailers should consider the effectiveness of tactile messaging. Tactile compensation videos influence consumers’ purchase intentions through perceived tactile diagnosticity; therefore, tactile design should focus, first and foremost, on improving perceptual diagnosticity. Designers should enhance the richness and effectiveness of a product’s tactile information [48]. Regarding tactile ‘reach’, retailers can add more cues to the interaction, such as character expressions, reactions, and the movement of natural scenery, to set the tactile mood. Other visual stimuli, such as color, shape, content, and familiarity, can be added as appropriate to adjust the attributes of the online screen to attract the user’s visual attention [82]. Additionally, high-quality interactions can increase consumers’ mental imagery and sensory similarity ratings, including human–product, object, and virtual scenario interactions [83]. Retailers can also attempt to increase sensory similarity ratings by inducing consumers to grasp objects with specific tactile characteristics within their reach as a substitute for conveying product-related tactile information. When presenting the tactile properties of products, it is vital to ensure that the information is authentic, the product presentation is clear, and the quality of the haptic cue interaction is high.

Second, retailers should account for the impact of their personal goals. For hedonic goal-oriented consumers, retailers should seek interesting ways to appeal to people’s emotions and design interesting programs. Indeed, when people pursue gaining goals, they experience hedonic feelings [35]. Thus, haptic cues should be designed to balance consumers’ hedonic and gainful pursuits. Effective measures should not only focus on increasing the perceived value to the consumer (e.g., product texture and tactile presentation) but also on increasing the relative appeal of multiple cues (e.g., making the tactile presentation of the product more humorous and lively). Simultaneously, consumers’ objectives should be inferred from the product’s features, functions, and elements designed accordingly. In this regard, even a product’s packaging may play an important role in influencing consumer product expectations and experiences of the product [13].

Finally, retailers should consider haptics strategically to improve customer relationships. Visual tactile compensation in an online environment can increase consumer valuation value, induce innovative positive reviews, and increase purchase intention [28,65]. Therefore, retailers must pay close attention to how the presented haptic cues interact with consumers; for example, by adding storytelling cues or demonstrations to videos that evoke essential details to improve consumers’ attitudes. Additionally, it explores tactile applications in online products that allow customers to interact with products or retailers more creatively. Finally, it provides enabling conditions for better evaluation, thus increasing customer recognition and understanding of the retailer’s capabilities, ultimately leading to positive reviews and behaviors.

## 6. Research Limitations and Recommendations

This study has some limitations. First, the data for this study were drawn from cross-sectional data. Therefore, richer data, including information on different products and longitudinal data, could be used in the future to ensure the generalizability of the findings. Second, the text discusses the role of personal goals in behavior. Individual heterogeneity is prevalent, and further opportunities lie in investigating the differences between consumer heterogeneity and consumption behavior. Third, because this study focuses on consumers’ attention to haptic cues—that is, customers’ attention to images and content—we exclude the interference of cues such as auditory cues (descriptions of language). A richer presentation video may contain more sensory cues, attracting consumers’ exogenous attention. However, the effects of exogenous attention are complex. Therefore, future research could focus on how multisensory interactions explain haptic properties in order to explore how and why different situations in which consumers activate or combine multisensory cues affect changes in perceived diagnosticity and purchase intentions.

## Figures and Tables

**Figure 1 behavsci-14-00050-f001:**
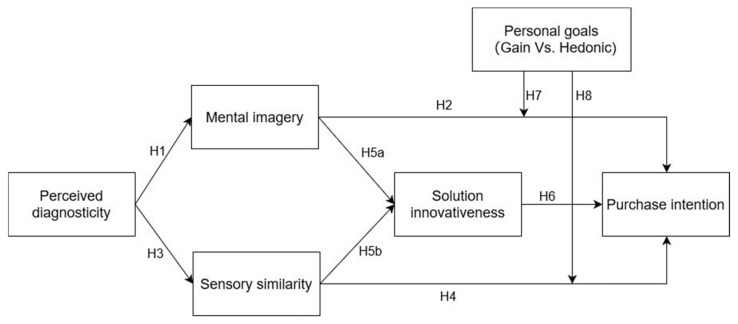
Conceptual model.

**Figure 2 behavsci-14-00050-f002:**
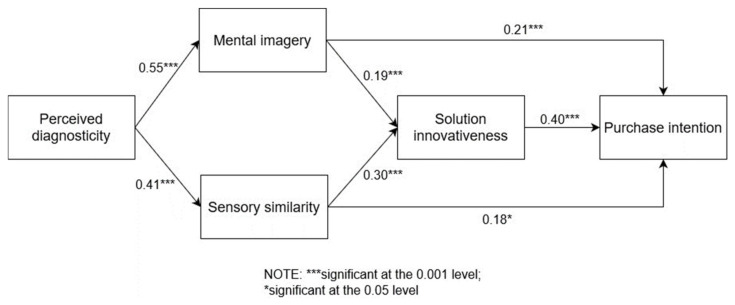
Results of the main effect.

**Figure 3 behavsci-14-00050-f003:**
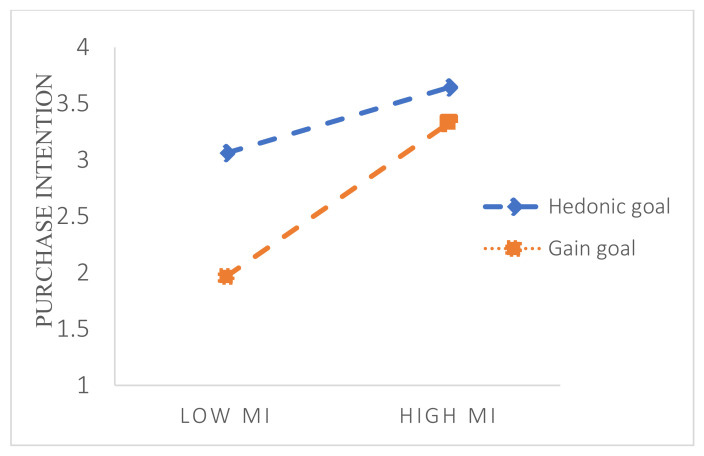
The effect of the interaction between mental imagery (MI) and personal goals on purchase intention.

**Table 1 behavsci-14-00050-t001:** Measurement items.

Constructs	Items	
Perceived diagnosticity [74]	PD1	The video demonstrating the feel or texture of the clothing helped me evaluate the tactile attributes of the product.
	PD2	The video demonstrating the feel or texture of the clothing helped familiarize me with the product.
	PD3	The video demonstrating the feel or texture of the clothing is helpful for me to understand the product’s tactile attributes.
Mental imagery [72]	MI1	During the video-viewing task, I imagined what it would be like to wear these clothes.
	MI2	During the video-viewing task, I fantasized about wearing the clothes.
	MI3	During the video-viewing task, I thought about what the feeling would be like when wearing the clothes.
	MI4	During the video-viewing task, I can easily imagine that I wear the clothes.
Sensory similar [27]	SS1	If I buy this product, I know I will have the same feeling as if I were inspecting the product directly.
	SS2	I feel that the product inspection in the video is similar to the direct product inspection.
	SS3	The video product shows the same feel as a direct product inspection.
Solutions innovativeness [61]	SI1	I find using product display videos to be advantageous in performing my shopping.
	SI2	The retailer offers a very convenient and useful way to display the products.
	SI3	The solutions offered by the retailer are novel.
	SI4	The retailer provides an unconventional way of solving problems.
Purchase intention [75]	PI1	After viewing this video, I became interested in making a purchase.
	PI2	After viewing this video, I am willing to purchase the product being presented.
	PI3	After viewing this video, I would consider purchasing the presented product.
	PI4	After viewing this video, I will likely buy this product.

**Table 2 behavsci-14-00050-t002:** Personal goals.

Goals	Sub-Goal	Potential Motives
Gain	Value for money	To receive value for money, pay a reasonable price, and avoid wasting money.
	Quality	To receive a high-quality and reliable product that meets personal expectations.
	Function	To receive a valuable and practical product that serves some purpose.
Hedonic	Pleasure	To receive something that satisfies an immediate need and makes one feel good and happy.
	Stimulating	To receive something exciting, stimulating, or unique, avoiding being boring.
	Comfortable	To receive something pleasant and comfortable, avoiding hassle and discomfort.

**Table 3 behavsci-14-00050-t003:** Demographic information of participants (*N* = 406).

Characteristics	Items	Frequency	Percentage (%)
Gender	Female	211	51.97
	Male	195	48.03
Age	<20	5	1.23
	21–30	183	45.07
	31–40	135	33.25
	41–50	73	17.99
	>50	10	2.46
Education	High school and below	21	5.17
	College students	27	6.65
	Undergraduate	319	78.57
	Postgraduate	39	9.61
Number of online purchases per month	Less than five times	132	32.5
	5–10 times	161	39.7
	More than ten times	113	27.8

**Table 4 behavsci-14-00050-t004:** Convergent validity.

Constructs	Items	Factor Loading	Cronbach’α	CR	AVE
Perceived diagnosticity	PD1	0.714	0.752	0.754	0.507
	PD2	0.756			
	PD3	0.662			
Mental imagery	MI1	0.753	0.835	0.835	0.558
	MI2	0.759			
	MI3	0.72			
	MI4	0.756			
Sensory similarity	SS1	0.703	0.761	0.761	0.515
	SS2	0.72			
	SS3	0.729			
Solution innovativeness	SI1	0.704	0.807	0.807	0.512
	SI2	0.762			
	SI3	0.719			
	SI4	0.674			
Purchase intention	PI1	0.776	0.868	0.869	0.624
	PI2	0.766			
	PI3	0.793			
	PI4	0.823			

**Table 5 behavsci-14-00050-t005:** Correlation coefficients and AVE values.

	PD	MI	SS	SI	PI
PD	0.712				
MI	0.427 ***	0.747			
SS	0.372 ***	0.385 ***	0.717		
SI	0.192 **	0.32 ***	0.366 ***	0.715	
PI	0.178 **	0.348 ***	0.326 ***	0.427 ***	0.79

Note: significant at *p*: ** ≤ 0.01; *** ≤ 0.001; diagonal lines represent square root values of AVE representing potential variables; PD = perceived diagnosticity; MI = mental image; SS = sensory similarity; SI = solutions innovativeness; PI = purchase intention.

**Table 6 behavsci-14-00050-t006:** Results of the main effect analysis.

Hypothetical Path		Estimate	S.E.	C.R.	P	Results
PD → MI	H1	0.554	0.083	6.645	***	Support
MI → PI	H2	0.214	0.061	3.492	***	Support
PD → SS	H3	0.406	0.071	5.737	***	Support
SS → PI	H4	0.182	0.079	2.287	0.022	Support
MI → SI	H5a	0.19	0.051	3.71	***	Support
SS → SI	H5b	0.299	0.067	4.452	***	Support
SI → PI	H6	0.397	0.081	4.901	***	Support

Note: *** *p* < 0.001. PD = perceived diagnosticity; MI = mental imagery; SS = sensory similarity; SI = solutions innovativeness; PI = purchase intention.

**Table 7 behavsci-14-00050-t007:** Results of the moderation effect analysis.

	Model 1			
	Coefficient	Standard Error	*t*	*p*
const	3.214	0.573	5.609	0.000 ***
Mental imagery	0.049	0.157	0.31	0.757
personal goals	−0.352	0.353	−0.998	0.319
int	0.195	0.096	2.026	0.043 **
R^2^	0.184			
R^2^ adjusted	0.178			
F	F(3, 402) = 30.267, *p* = 0.000 ***			

Note: *** *p* < 0.001, ** *p* < 0.05;

**Table 8 behavsci-14-00050-t008:** Results of the moderation effect analysis.

	Model 2			
	Coefficient	Standard Error	*t*	*p*
const	3	0.537	5.587	0.000 ***
Sensory similarity	0.201	0.147	1.367	0.172
Personal goals	−0.041	0.331	−0.124	0.901
Int	0.041	0.09	0.455	0.65
R^2^	0.089			
R^2^ adjusted	0.082			
F	F(3, 402) = 13.091, *p* = 0.000 ***			

Note: *** *p* < 0.001;

## Data Availability

The data supporting the findings of this study are available from the corresponding author upon reasonable request.

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
