# Peer review of "Seeing as Feeling? The Impact of Tactile Compensation Videos on Consumer Purchase Intention"

_behavsci, 2024, doi:10.3390/bs14010050_

Round 1

Reviewer 1 Report

Comments and Suggestions for Authors

In my opinion, the article meets the quality criteria for scientific research.

All the sections are adequately carried out, both the introduction, the hypothesis statement, the structural equation model and the analysis of the study.

The only thing that could be corrected is the approach of the headings within the methodology, after the pretest I would see more appropriate to specify two sections, one for data collection and the other for data analysis.

The discussion and conclusions are also adequately carried out.

Author Response

We feel great thanks for your professional review work on our article. There is already a separate section on data analysis in the thesis (Section 4. Data Analysis and Results). According to your nice suggestions, I will add Data collection to Section 3. The changed structure is as follows. "

  1. Methodology

3.1. Measurement

3.2. Pretest

3.3. Materials and Data Collection

Reviewer 2 Report

Comments and Suggestions for Authors

The topic is actually, and attention-grabbing. The title is too long that is why I suggest deleting the second part of it because, among the keywords, both terms will be repeated.  

Formal recommendation: The authors placed reference numbers not only in square brackets but also in superscripts. Please, remove the superscript. 

Overall, the content is logical and delivers new insights into consumer research.  Both theoretical and practical implementation of results is important.

Reviewer 3 Report

Comments and Suggestions for Authors

This submission has a potential to provide further understanding to both academics and practitioners interested in explaining the role of tactile perceived diagnosticity on consumer purchase intention. It is obvious that this study is going to practically and theoretically contribute to understanding technology-based social entrepreneurs and related areas. However, some comments need to be incorporated.  

 The introduction section:

·       Few paragraphs started with transition words. The language should be reviewed as new paragraphs should not start with a transition word. Example, Line 49 starts with However and line 76 starts with Furthermore.

·       The three research questions presented in the last paragraph in the introduction. ((a) The role of visually compensated tactile perceptual diagnosticity on mental imagery, sensory similarity, and consumer purchase intention. (b) Can programs that provide effective haptic cues in videos serve as a basis for retailers' innovativeness judgments? (c) The moderating role of personal goals). Only (b) is a question, (a) and (c) should be revised to be questions.

·       One additional paragraph should be added in the introduction at the end to present the coming sections of the paper.

 The title of Figure 1.  should be revised to (Conceptual model) not (Structural model)

The results should be presented in (Figure 2.  Structural model) in section 4. Model Analysis and Results

 The conclusion should be the last section of the paper as following:

 5. Discussion

5.1. Theoretical Implications

5.2. Management Insights

6. Research Limitations and Recommendations

7. Conclusions

·       More up-to-date related studies should be cited to support this submission especially in 2023. 

·       The language of the paper needs a careful editing. 

Comments on the Quality of English Language

The language of the paper needs a careful editing. 

Reviewer 4 Report

Comments and Suggestions for Authors

Overall comments

The authors have done a remarkable job by conducting research on such an important subject that has potential to impact a very large population around the world. The manuscript contributes immensely to the current research on the role of tactile demonstration video effect on consumers purchase decision. However, the language used in the manuscript is too complex and technical. The heavy use of technical terms such as haptic, tactile, perceived diagnosticity, mental imagery, etc., have made it almost impossible to follow the storyline and it sounded very repetitive. Authors are encouraged to revise the language by making it a simple read manuscript. The purpose of publishing a research manuscript is to disseminate knowledge rather than limiting it due to complex language.

Please see the specific comments below.

·         Line 13-14: “intention based … Chinese markets”. Please rewrite these sentences.

·         Line 37-40: “One approach … the product”. Please provide examples for each approach for reader to relate and understand them correctly.

·         Line 73-74: “the pleasantness .. information”. Please cite the source of this information.

·         Table 1: The language used in table 1 is very technical, especially terms such as “tactile”.  How did authors ensure that consumers were able to understand the terms in the way it was intended.

·         Were there any criteria used that involved garment and fabrics usage? Such as type of garment/fabrics participants often buy online, frequent users of a particular garment/fabrics, etc.

·         How did the authors ensure that they are selecting right participants who can evaluate chiffon dresses and cotton shirt online?

Comments on the Quality of English Language

 The language used in the manuscript is too complex and technical. The heavy use of technical terms such as haptic, tactile, perceived diagnosticity, mental imagery, etc., have made it almost impossible to follow the storyline and it sounded very repetitive. Authors are encouraged to revise the language by making it a simple read manuscript. The purpose of publishing a research manuscript is to disseminate knowledge rather than limiting it due to complex language.

Round 2

Reviewer 4 Report

Comments and Suggestions for Authors

All comments are addressed. No further suggestions for improvement.